# Outcome Using Either Intradermal Botox Injection or Endoscopic Thoracic Sympathectomy for Patients with Primary Palmar Hyperhidrosis: A Comparative Study

**Ghadah Alhetheli**

Department of Dermatology, Faculty of Medicine, Qassim University, Qassim 56218, Saudi Arabia; ghthly@qu.edu.sa

**Abstract:** Hyperhidrosis, or excessive sweating, negatively impacts patients both physically and psychologically. It may be primary or secondary: the primary form is a benign condition, with its growing prevalence reaching 5% recently. Its medical treatments are transitory. Objectives: Comparison of the outcomes of patients with primary palmar hyperhidrosis (PPH) after intradermal Botox injection (IBI) versus endoscopic thoracic sympathectomy (ETS). Methods: Forty patients were randomly divided into two equal groups. Patients in the IBI group received an intradermal injection of a botulinum toxin A. Patients in the EST group received endoscopic electrocautery of the sympathetic chain. The patients were evaluated biweekly for 12 weeks, and patient satisfaction by outcome was evaluated using a 4-point satisfaction score. Results: At 12 weeks, 60% of the IBI group patients had maintained an improvement. Meanwhile, 40% of the patients were improved compared to pre-intervention scores, despite deterioration after remarkable improvement. On the other hand, 80% of ETS group patients maintained their Hyperhidrosis Disease Severity Scale (HDSS) up until the end of follow-up. Patient satisfaction scores were significantly higher for the IBI group compared to the ETS group. Conclusions: Intradermal Botox injection is a simple, safe, non-invasive, and effective therapeutic modality for PPH and achieved higher patient satisfaction compared to ETS.

**Keywords:** palmar hyperhidrosis; Botox therapy; surgical intervention; outcome; satisfaction

## 1. Introduction

Primary focal hyperhidrosis has a deleterious effect on patients' quality of life with growing prevalence reaching 5% in recent literature [1]. Primary palmar hyperhidrosis (PPH) is abnormal over-sweating of the palms usually associated with hyperhidrosis elsewhere in the body, especially plantar hyperhidrosis [2]. PPH is a chronic neurologic disorder characterized by excessive sweating of the eccrine glands due to sympathetic overactivity [3]. Acetylcholine is the neurotransmitter used in the somatic nervous system, the preganglionic and postganglionic fibers of the parasympathetic nerves, and the preganglionic fibers or postganglionic sympathetic nerves [4].

Botulinum neurotoxin (BoNT), produced by *Clostridium botulinum*, is the most potent toxin; [5] it is a zinc protease that acts through the cleavage of neuronal vesicle-associated proteins responsible for the release of acetylcholine into the neuromuscular junction [6]. While there are seven biological serotypes of BoNT, only two are used in clinical practice [7]: BoNT serotypes A and B are used for a variety of medical indications, including facial wrinkles, dystonia, spasticity, migraines, multiple sclerosis, post-stroke gait disorders, and cerebral palsy deformities [8–11].

Despite the presence of effective and minimally invasive curative treatments, repetitive, non-curative symptomatic strategies dominate the current treatment of primary hyperhidrosis [1]. Considering that the neuropathogenesis of PPH is increased sympathetic cholinergic sudomotor nerve traffic to the palmar surface of the hands [3,12], therapeutic interventions that cause either the interruption of sympathetic flow to the upper extremity

through surgical [13] or ablative sympathectomy [14] or local injection of botulinum toxin A are very effective, especially for cases resistant to conservative therapy [15]. The study sought to define and compare the short-term outcome of managing PPH using either intradermal Botox injection (IBI) or endoscopic thoracic sympathectomy (ETS) and evaluate patient satisfaction by the outcome.

## 2. Materials and Methods

### 2.1. Design

This prospective interventional comparative study aimed to determine the short-term outcome of managing PPH using either intradermal Botox injection (IBI) or endoscopic thoracic sympathectomy (ETS) and evaluate patient satisfaction by the outcome. The study was conducted following the guidelines of the Declaration of Helsinki and approved by Local Ethical Committee (RC: 10-3-19). In conjunction between Dermatology Department, Faculty of Medicine, Qassim University, Saudi Arabia and multiple private centers in Egypt, the study intended to include all patients attending with a complaint of palmar hyperhidrosis (PH) from March until May of 2019. Written informed consent was obtained from all the subjects before enrollment and after an explanation of the aim and nature of the study. All patients were subjected to clinical examination to establish a diagnosis with PH and history taking concerning previous management lines and their results.

### 2.2. Evaluation of PH Disease Severity

Disease severity was determined using the Hyperhidrosis Disease Severity Scale (HDSS), which is a validated subjective survey used to characterize disease severity on a 4-grade scale: mild disease is scored as 1, moderate disease is scored as 2, and severe disease is scored as 3 or 4 according to the extent of interference with daily activities. The HDSS score was determined at the time of the patient's enrollment in the study and biweekly for 3 months; a 1-point improvement indicated a 50% reduction of sweat production and a 2-point improvement indicated an 80% reduction of sweat production [16,17].

Disease severity was also determined by objective evaluation: the hands were dried and re-produced sweat was then observed. Sweating severity was evaluated using a 6-point visual scale [18] as shown in (Figure 1) to determine the amount of sweat produced.

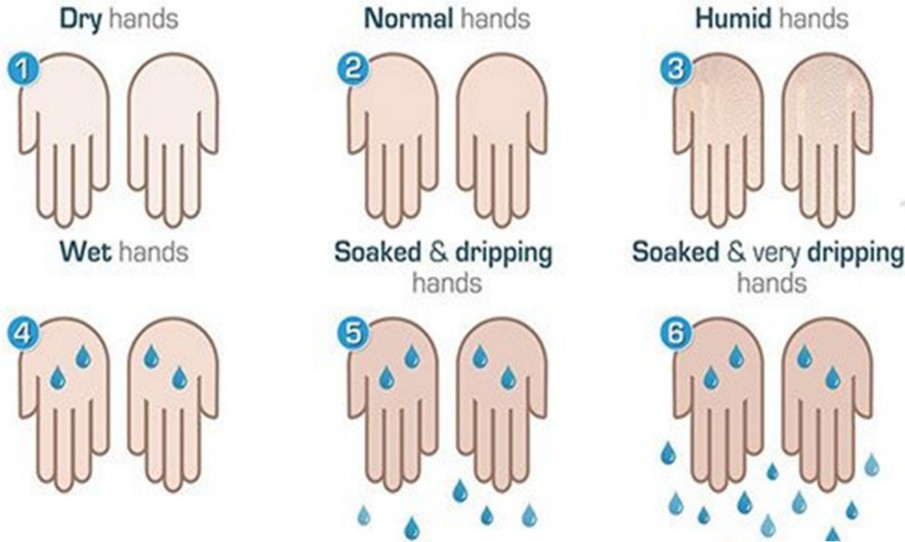

**Figure 1.** Visual scale for the quantification of palmar hyperhidrosis. Source: Lyra [18].

### 2.3. Localization of Sweaty Areas

Minor's starch–iodine test was used to determine the precise area for treatment. Iodine solution (3–5%) was applied to a dried hand and allowed to dry. Then, starch was applied,

which turned purple when sweating occurred in contact with the iodine and starch [19]. Areas of excessive sweating were demarcated on both hands and as a reference on drawings of the human body.

### 2.4. Exclusion Criteria

Exclusion criteria included pregnancy, breastfeeding, and systemic diseases related to PH; previous or refusal to have a sympathetic blockade and/or Botox injection; coagulopathy; hepatic diseases and refusal to participate in the trial, sign the written consent, or attend scheduled follow-up visits.

### 2.5. Inclusion Criteria

Patients with PH who failed to respond to conservative medical treatment and agreed to undergo either a thoracic sympathectomy or Botox injection were included in the study.

### 2.6. Randomization and Grouping

Patients were randomly allocated into one of two study groups: a Botox group that included patients assigned for intradermal Botox injection (IBI group) and a thoracic sympathectomy group that included patients assigned to undergo endoscopic thoracic sympathectomy (ETS group). Randomization was achieved using opaque envelopes containing cards carrying the group label, which were prepared by an assistant who was blinded to the significance of the labels (Figure 2).

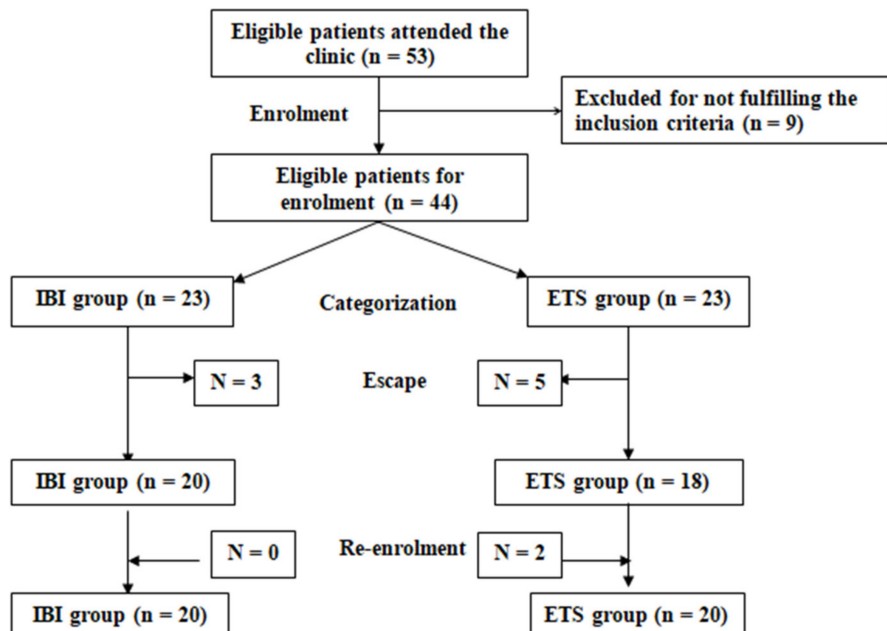

**Figure 2.** Study Flow Chart.

### 2.7. Study Procedures

#### 2.7.1. IBI Group

After identification of the sweaty areas using Minor's starch–iodine test and sterilization of the palm using Povidone iodine solution, the palm was rubbed with xylocaine gel and divided into 1.5 × 1.5 cm squares. Two points were identified on the medial and lateral edges of the phalanges, and a third point was marked on the tip of the distal phalanx. A 50 IU vial of on a botulinum toxin A (BOTOX; Allergan Inc., Madison, NJ, USA), reconstituted using 10 mL of sterile normal saline solution, was used in a concentration of 0.5-unit/0.1 mL, which was previously documented to minimize the effect of the toxin on the handgrip [12,20]. The Botox was injected using a 27-gauge needle with its bevel

directed upwards and introduced at an angle of 45o to the surface into the dermis nearest to the subcutaneous tissue but not crossing it. The injections were conducted as described previously [21] at predetermined points at a dose of 0.1 mL/injection site. After the injections, firm pressure without massage was applied to the injection site to aid hemostasis, prevent bruising, and allow drug distribution. At the end of the procedure, the hand was closed tightly, and a crepe bandage was applied for 2 h to keep the hand closed to allow fixation of the toxin to nerve endings [16].

### 2.7.2. ETS Group

All surgeries were conducted under general anesthesia with endotracheal intubation as previously described by Kuijpers et al. [22] to achieve endoscopic electrocautery of the sympathetic chain on the surface of the 3rd rib or the 3rd to the 5th rib, and division was extended 2 cm laterally over the rib to transect accessory nerve fibers.

### 2.7.3. Follow-up

The patients were followed up biweekly for 3 months for HDSS evaluation, and patient satisfaction by outcome was evaluated using a 4-point satisfaction score. The follow-up evaluations were performed by an assistant, who was not included as an author of this paper and had previously performed pre-interventional evaluations.

### *2.8. Study Outcomes*

The study outcomes included the extent of hand dryness as scored by the HDSS, development of regression after improvement, development of complications, and patient satisfaction by outcome.

### 3. Statistical Analyses

The obtained data are presented as mean, standard deviation (SD), numbers, percentages, medians and interquartile range. The results were analyzed using the one-way ANOVA test for the parametric results and the Mann–Whitney U test and the chi-squared test for the non-parametric results. The statistical analyses were conducted using the IBM SPSS (Version 23, 2015; IBM, South Wacker Drive, Chicago, IL, USA) for Windows statistical package. A *p*-value < 0.05 was considered statistically significant.

### 4. Results

The study included 57 patients presenting with PPH; 17 were excluded for not fulfilling the inclusion criteria, and the remaining 40 were randomly divided into two equal groups. The demographic and clinical data collected at the time of enrollment were not significantly different (Table 1).

Time-scale improvement of individual patients according to HDSS score are shown in Table 2 with a reference to the percentage of score change at the 12th versus the initial score at 0 week. As shown in Table 2, HDSS of IBI group showed varied scores during the 12-week follow-up period, while the ETS group developed gradual change of their HDSS until the 4th week, and this score remained stationary until the 12th week of follow-up. The median values of the percentage of change of HDSS score at end of follow-up versus at start was 50 (IQR: 50–80) for the IBI group and 80 (IQR: 50–80) for the ETS group with non-significantly higher median value for percentage of change of HDSS with the ETS group versus the IBI group.

**Table 1.** Patients' demographic and clinical data.

| | | | IBI Group | ETS Group | *p*-Value |
|---|---|---|---|---|---|
| Age (Years) | Categories | <20 | 2 (10%) | 0 | 0.791 |
| | | 20–25 | 8 (40%) | 12 (60%) | |
| | | 26–30 | 6 (30%) | 5 (25%) | |
| | | >30 | 4 (20%) | 3 (15%) | |
| | Mean (SD) | | 25.4 (4.7) | 26.7 (5.1) | 0.456 |
| | Range | | 19–35 | 22–41 | |
| Sex | Males | | 9 (45%) | 8 (40%) | 0.749 |
| | Females | | 11 (55%) | 12 (60%) | |
| Body Mass Index (kg/m$^2$) | Categories | <30 | 12 (60%) | 11 (55%) | 0.749 |
| | | 30–35 | 8 (40%) | 9 (45%) | |
| | Mean (SD) | | 29.7 (1.7) | 29.7 (2.2) | 0.955 |
| | Range | | 26.7–33 | 26.6–33 | |
| HDSS | Categories | Grade 1 | 0 | 0 | 0.355 |
| | | Grade 2 | 4 (20%) | 1 (5%) | |
| | | Grade 3 | 9 (45%) | 11 (55%) | |
| | | Grade 4 | 7 (35%) | 8 (40%) | |
| | Median | | 3 | 3 | 0.465 |
| | IQR | | 3–4 | 3–4 | |
| Objective Evaluation | Categories | Humid (score = 3) | 2 (10%) | 0 | 0.898 |
| | | Wet (score = 4) | 6 (30%) | 8 (40%) | |
| | | Soaked and dripping (score = 5) | 8 (40%) | 8 (40%) | |
| | | Soaked and very dripping (score = 6) | 4 (20%) | 4 (20%) | |
| | Median | | 5 | 5 | 0.841 |
| | IQR | | 4–5 | 4–5 | |

Data are presented as mean, standard deviation (SD), range, median, and interquartile range (IQR). IBI: intradermal Botox injection; ETS: endoscopic thoracic sympathectomy; HDSS: Hyperhidrosis Disease Severity Scale; *p*-value indicates the significance of the difference between both groups; $p < 0.05$ indicates significant difference; and $p > 0.05$ indicates a non-significant difference.

**Table 2.** Individual HDSS scores and the percentage score change at end of the 12-week follow-up.

| | IBI Group | | | | | | | | ETS Group | | | |
|---|---|---|---|---|---|---|---|---|---|---|---|---|
| No. | 0-wk | 2-wk | 4-wk | 6-wk | 8-wk | 10-wk | 12-wk | % | No. | 0-wk | 2-wk | 4-wk | % |
| 1 | 2 | 1 | 1 | 1 | 1 | 1 | 1 | 50 | 1 | 4 | 2 | 2 | 80 |
| 2 | 3 | 2 | 1 | 1 | 1 | 1 | 1 | 80 | 2 | 3 | 1 | 1 | 80 |
| 3 | 4 | 3 | 2 | 2 | 2 | 2 | 2 | 80 | 3 | 4 | 2 | 2 | 80 |
| 4 | 3 | 1 | 1 | 1 | 2 | 2 | 2 | 50 | 4 | 3 | 1 | 1 | 80 |
| 5 | 2 | 1 | 1 | 1 | 1 | 1 | 1 | 50 | 5 | 3 | 1 | 1 | 80 |
| 6 | 4 | 2 | 2 | 2 | 2 | 2 | 2 | 80 | 6 | 4 | 2 | 2 | 80 |
| 7 | 3 | 1 | 1 | 1 | 1 | 1 | 1 | 80 | 7 | 3 | 1 | 1 | 80 |
| 8 | 4 | 3 | 2 | 2 | 3 | 3 | 3 | 50 | 8 | 4 | 2 | 2 | 80 |
| 9 | 2 | 1 | 1 | 1 | 1 | 1 | 1 | 50 | 9 | 2 | 1 | 1 | 50 |
| 10 | 3 | 2 | 2 | 1 | 1 | 1 | 1 | 80 | 10 | 3 | 1 | 1 | 80 |
| 11 | 4 | 3 | 2 | 2 | 2 | 2 | 3 | 50 | 11 | 4 | 2 | 2 | 80 |

**Table 2.** *Cont.*

| IBI Group | | | | | | | | | ETS Group | | | | |
|---|---|---|---|---|---|---|---|---|---|---|---|---|---|
| 12 | 3 | 2 | 2 | 1 | 1 | 2 | 2 | 50 | 12 | 3 | 1 | 1 | 80 |
| 13 | 4 | 3 | 2 | 2 | 2 | 2 | 2 | 80 | 13 | 4 | 2 | 2 | 80 |
| 14 | 3 | 2 | 1 | 1 | 1 | 1 | 1 | 80 | 14 | 3 | 2 | 1 | 80 |
| 15 | 2 | 1 | 1 | 1 | 1 | 1 | 1 | 50 | 15 | 3 | 1 | 1 | 80 |
| 16 | 4 | 2 | 2 | 2 | 2 | 2 | 2 | 80 | 16 | 4 | 2 | 3 | 50 |
| 17 | 3 | 2 | 2 | 1 | 1 | 2 | 2 | 50 | 17 | 3 | 2 | 2 | 50 |
| 18 | 4 | 3 | 3 | 2 | 2 | 3 | 3 | 50 | 18 | 4 | 4 | 4 | 0 |
| 19 | 3 | 2 | 1 | 1 | 2 | 2 | 2 | 50 | 19 | 3 | 3 | 3 | 0 |
| 20 | 3 | 3 | 2 | 2 | 2 | 2 | 2 | 50 | 20 | 3 | 2 | 2 | 50 |
| Median | 3 | 2 | 2 | 1 | 1.5 | 2 | 2 | 50 | | 3 | 2 | 2 | 80 |
| IQR | 3–4 | 1–3 | 1–2 | 1–2 | 1—2 | 1–2 | 1–2 | 50–80 | | 3–4 | 1–2 | 1–2 | 50–80 |
| *p* value | 0.456 | 0.298 | 0.764 | 0.263 | 0.596 | 0.968 | 0.857 | | | | | | |
| P1 | | 0.0003 | <0.0001 | <0.0001 | <0.0001 | <0.0001 | <0.0001 | | | | <0.0001 | <0.0001 | 0.201 |

IBI: intradermal Botox injection; ETS: endoscopic thoracic sympathectomy; %: percentage of the Hyperhidrosis Disease Severity Scale (HDSS) score change determined at the 12-week follow-up vs. the score determined during the pre-interventional; *p*-values indicate the significance of the difference between the groups; P1 indicates the significance between the scores determined at each visit vs. the pre-interventional score; $p < 0.05$ indicates a significant difference; $p > 0.05$ indicates a non-significant difference; and interquartile range (IQR).

Among the IBI group, four (20%) patients improved by 50% and three (15%) patients improved by 80% at the 2nd week of follow-up, and their improvement was maintained until the 12-week of follow-up. Another five (25%) patients were improved by 80% at the 4th week of follow-up, and their improvement was maintained until the end of the follow-up. The remaining eight patients showed 1–2 point improvements on the HDSS scale, but they later worsened, despite a final improvement by 50% in comparison to the initial pre-injection scorings. For the ETS group, at the 2nd week of follow-up, four (20%) patients were improved by 50%, 14 (70%) patients were improved by 80%, but two (10%) patients showed no improvement. At the 4-week follow-up, two patients showed a 1-point regression, while the remaining 16 patients maintained their HDSS until the end of the follow-up period (Table 2).

The distribution of patients of both groups according to HDSS grades during the follow-up period showed no significant differences (Table 3).

**Table 3.** Distribution of patients among HDSS grades as determined during the 12-week follow-up period.

| Time Group | Baseline | | 2-wk | | 4-wk | | 6-wk | | 8-wk | | 10-wk | | 12-wk | |
|---|---|---|---|---|---|---|---|---|---|---|---|---|---|---|
| | IBI | ETS | IBI | ETS | IBI | ETS | IBI | ETS | IBI | ETS | IBI | ETS | IBI | ETS |
| Grade 0 | 0 | 0 | 6 | 8 | 9 | 9 | 12 | 9 | 10 | 9 | 8 | 9 | 8 | 9 |
| Grade 1 | 4 | 1 | 8 | 10 | 10 | 8 | 8 | 8 | 9 | 8 | 10 | 8 | 9 | 8 |
| Grade 2 | 9 | 11 | 6 | 1 | 1 | 2 | 0 | 2 | 1 | 2 | 2 | 2 | 3 | 2 |
| Grade 3 | 7 | 8 | 0 | 1 | 0 | 1 | 0 | 1 | 0 | 1 | 0 | 1 | 0 | 1 |
| *p*-Value | 0.356 | | 0.285 | | 0.911 | | 0.881 | | 0.936 | | 0.999 | | 0.961 | |

IBI: intradermal Botox injection; ETS: endoscopic thoracic sympathectomy; *p*-value indicates the significance of the difference between the groups; and $p > 0.05$ indicates a non-significant difference.

Objectively, the pre-IBI evaluation showed that four patients had soaked and very dripping hands, eight patients had soaked and dripping hands, six patients had soaked hands, and only two patients had wet hands. At the 2-week follow-up, 16 patients had normal hands, while three of those who had very dripping hands developed wet hands. At the 4-week follow-up, 19 patients had normal hands free of compensatory hyperhidrosis, and these evaluations were maintained until the end of the follow-up period, as assessed

by Minor's starch–iodine test. Only one patient still had soaked hands for a frequency of compensatory hyperhidrosis after IBI of 5%. For the patients who received ETS, four patients had soaked and very dripping hands, eight patients had soaked and dripping hands, and eight patients had soaked hands. At the 2-week follow-up, two patients showed no change, three patients had soaked hands, and 14 patients had normal hands, but unfortunately, one patient developed dry hands. These findings were continued up to 4 weeks postoperatively. Collectively, at the end of follow-up, only one patient in the IBI group had compensatory hyperhidrosis versus 6 patients in the ETS group; 4 had compensatory hyperhidrosis, and two had complete failure of the procedure (Table 4) (Figure 3).

**Table 4.** Objective evaluation of patients up to the 4-week follow-up.

| Time Objective Type Group | Baseline | | 2-wk | | 4-wk | |
|---|---|---|---|---|---|---|
| | IBI | ETS | IBI | ETS | IBI | ETS |
| Dry (score = 1) | 0 | 0 | 0 | 1 (5%) | 0 | 1 (5%) |
| Normal (score = 2) | 0 | 0 | 16 (80%) | 14 (70%) | 19 (95%) | 14 (70%) |
| Humid (score = 3) | 2 (10%) | 0 | 4 (20%) | 3 (15%) | 1 (5%) | 3 (15%) |
| Wet (score = 4) | 6 (30%) | 8 (40%) | 0 | 0 | 0 | 0 |
| Soaked and Dripping (score = 5) | 8 (40%) | 8 (40%) | 0 | 1 (5%) | 0 | 1 (5%) |
| Soaked and Very Dripping (score = 6) | 4 (20%) | 4 (20%) | 0 | 1 (5%) | 0 | 1 (5%) |
| *p*-Value | 0.938 | | 0.461 | | 0.191 | |

IBI: intradermal Botox injection; ETS: endoscopic thoracic sympathectomy; *p*-value indicates the significance of the difference between the groups; and *p* > 0.05 indicates a non-significant difference.

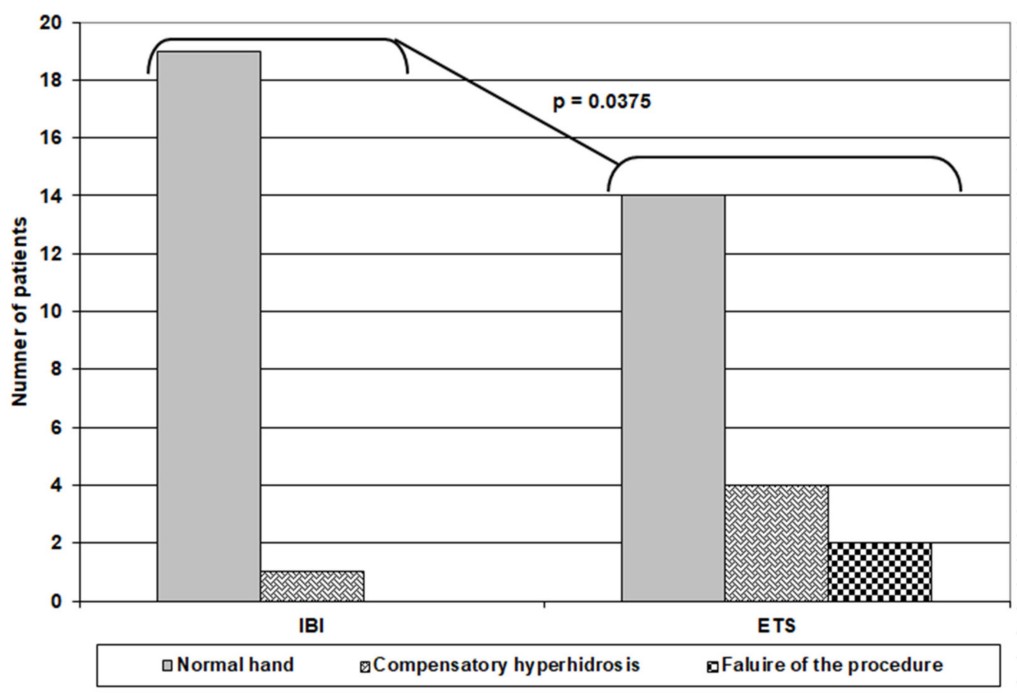

**Figure 3.** Outcome of patients of both groups at the end of 12-week follow up.

Only one patient in the IBI group developed weakness of the left-hand intrinsic muscles, which resolved after 4 months. However, all patients who received IBI complained of pain during the injection, and this was exaggerated in the females. Out of the patients who received ETS, three cases required a hospital stay for more than 48 h for decreased

oxygen saturation, and one patient developed a wound infection. Fortunately, these four patients responded to conservative medical treatment and were discharged.

Regarding patients' satisfaction, 24 patients were very satisfied with the outcome, eight patients were satisfied, four patients were dissatisfied, and four patients in the ETS group were very dissatisfied, with a significantly higher number of very satisfied patients in the IBI group (Table 5) (Figure 4).

**Table 5.** Patients' satisfaction by management outcome.

| Satisfaction Score | IBI | ETS | *p*-Value |
|---|---|---|---|
| Very Dissatisfied | 0 | 4 (20%) | |
| Dissatisfied | 2 (10%) | 2 (10%) | |
| Satisfied | 3 (15%) | 5 (25%) | 0.032 |
| Very Satisfied | 15 (75%) | 9 (45%) | |

IBI: intradermal Botox injection; ETS: endoscopic thoracic sympathectomy; *p*-value indicates the significance of the difference between the groups; and $p < 0.05$ indicates a significant difference.

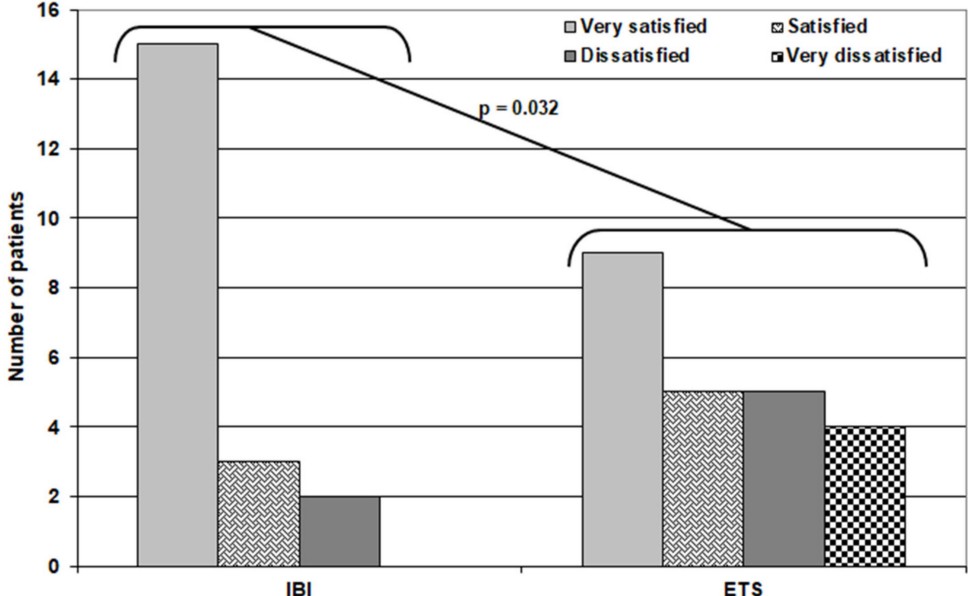

**Figure 4.** Distribution of patients of both groups according to their satisfaction score.

## 5. Discussion

This study compared the outcome of managing patients with PPH by either IBI or ETS; IBI maintained improvement for 12 weeks in 60% of patients, and 40% of patients showed an improved HDSS at 12 weeks compared to their baseline score despite deterioration after remarkable improvement. These findings illustrate the efficacy of IBI, considering that all patients had improved up until the 8-week follow-up and 60% maintained this improvement.

Similarly, Campanati et al. [23] reported a significant decrease in PPH severity after Botox injections at a 4-week follow-up. Bernhard et al. [24] also used 75–100 U per sole Botox injections for plantar hyperhidrosis in pediatric patients and reported a good therapeutic effect of variable duration in most patients with no severe side effects. Mirkovic et al. [25] reported the complete disappearance of sweating in 91.2% of their patients who received Botox injections for the management of focal and multifocal hyperhidrosis in children.

Recently, Agamia et al. [26] found that the delivery of 75 U of Botox to the palm dermis assisted by fractional $CO_2$ laser was clinically equivalent to intradermal injection of 50 U and significantly reduced pain, so this can be considered an effective and safe

alternative for the treatment of PPH with minimal side effects and complications. De Almeida et al. [27] reported a significant ≥ 50% sweat reduction from the baseline after 30 days of treatment with 90%–100% reduction of sweating by Minor's starch–iodine test after multiple puncture injections of botulinum toxin, and this response was sustained for at least 270 days.

The only drawback of IBI was pain during the intradermal injection despite the pre-emptive application of local anesthetic cream; unfortunately, its absorption was insufficient to prevent injection pain, which was manifested more by females. Similarly, Bernhard et al. [24] reported that 60% of their patients experienced injection site pain that lasted for 3 days. However, the pain of an intradermal Botox injection remains a problem, and multiple recent trials have tried to provide solutions using modified injection techniques [26,28] or manipulations of the pH of the solution, [29] which is the main cause of pain initiation.

Only one patient (5%) complained of mild weakness of the left-hand intrinsic muscles post-IBI. Despite this complication being previously documented as common for Botox injections, [23,30] it was minimized in the current study's patients due to avoiding piercing the dermal–subcutaneous interface during the injections to spare the motor nerves, avoiding massage after injection, asking the patients to lightly close their hands to allow the minimal spread of the toxin, and the use of the smallest effective dose (i.e., 50 U). The patient reported regaining the power of his left-hand muscles over the 4-month follow-up.

In line with the obtained results and the applied precautions, Schnider et al. [30] reported significant subjective and objective scorings for PPH until 13 weeks after injection and reported muscle weakness, which lasted for 2–5 weeks in 27% of their patients. On the contrary, Saadia et al. [12] documented that handgrip strength was not affected by Botox for the management of PPH, and the reduction of finger pinch strength was significantly reduced by the injection of 50 U compared to 100 U of Botox. Bernhard et al. [24] reported transient focal weakness for 4 weeks in 6.7% of their pediatric patients who received a Botox injection for plantar hyperhidrosis.

Endoscopic thoracic sympathectomy also significantly reduced sweating severity in comparison to baseline scores, and the reported outcomes were stationary up until the 4-week follow-up. However, the reduction in HDSS scores was not significantly higher compared to corresponding figures obtained by IBI. Unfortunately, two cases failed to respond (10% failure rate), another three cases required a hospital stay of more than 48 h for decreased oxygen saturation, and one patient developed a wound infection that responded to conservative medical treatment. These results point to a finding that despite the sustained high reduction rate of sweating, the procedure carries many limitations, including being an invasive technique and the possibility of failure and postoperative complications. These results go in hand with multiple recent studies that have evaluated the outcomes of thoracic sympathectomy. Kuijpers et al., [1] using a bilateral one-stage single-port sympathectomy procedure, reported not only a significant reduction in HDSS scores but also severe compensatory hyperhidrosis with a frequency ranging between 17.1% and 32.8% of patients. In a comparative study, Andresen et al. [31] compared CT-guided thoracic sympathicolysis and video-assisted thoracoscopic sympathectomy for PPH and found both treatments resulted in a marked reduction of symptoms, but transient compensatory sweating was reported in 14.9% of cases after both approaches, and video-assisted thoracoscopic sympathectomy resulted in postoperative pneumothorax in 8% and temporary thoracotomy pain in 14% of patients. In another comparative study of T3 thoracoscopic sympathectomy versus T3+T4 sympathectomy, Yang et al. [32] found that both techniques resulted in a significant reduction of a sweating index, but they also resulted in compensatory hyperhidrosis with an incidence of 5–10%, with the possibility of postoperative neuralgia and pneumothorax requiring chest tube placement.

In support of the efficacy of Botox injection over ETS as a therapeutic modality for hyperhidrosis, Park et al. [33] documented that Botox injection into the stellate ganglion is simple and safe and produces longer-lasting effects for craniofacial hyperhidrosis than endoscopic sympathectomy and a single nerve block.

**Limitations**: The small study size and short duration of the follow-up are the study limitations.

**Recommendations**: A comparative study using the recent modification to reduce injection pain is advocated.

## 6. Conclusions

Intradermal Botox injection is a safe, effective, and minimally invasive therapeutic modality for PPH with a comparable outcome to thoracic sympathectomy. Intradermal Botox injection has higher patients' satisfaction compared to ETS. Injection pain is the most frequent side effect of IBI, and intrinsic hand muscle weakness is infrequent and reversible.

**Funding:** This research received no external funding.

**Institutional Review Board Statement:** The study was conducted according to the guidelines of the Declaration of Helsinki, and approved by the Research Ethical committee (REC: 10-3-19).

**Informed Consent Statement:** Written informed consent was obtained from all the subjects before enrollment and after an explanation of the aim and nature of the study.

**Data Availability Statement:** Not applicable.

**Acknowledgments:** The author thanks staff members of the vascular surgery team at Tanta University Hospital and dermatologists at Glory Medical Center at Tanta, Egypt for their effective sharing in this work.

**Conflicts of Interest:** The author declares no conflict of interest.

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
