# Peer review of "Outcome Using Either Intradermal Botox Injection or Endoscopic Thoracic Sympathectomy for Patients with Primary Palmar Hyperhidrosis: A Comparative Study"

_cosmetics, doi:10.3390/cosmetics8020041_

Round 1
Reviewer 1 Report
An interesting original study comparing the effectiveness of Intradermal Botox Injection with Endoscopic Thoracic Sympathectomy in palmar hyperhidrosis, showing higher satisfaction score in patients treated with botox injections; only minor queries:
The exclusion criteria should be broadened; for example, I do believe that pregnancy or previous injections of botox may enter in these criteria.
Given the high invasiveness of Endoscopic Thoracic Sympathectomy, I would have probably not have created a study with 2 equal groups.
In the results, the 2 groups' results should be more easily compared; the paragraph should be changed accordingly
Thank You
Thank You
Author Response
"Please see the attachment."

Reviewer 2 Report
This is a prospective randomized comparative study evaluating the effectiveness of primary palmar hyperhidrosis correction with BOTOX injection and thoracoscopic sympathotomy.
The results suggest that overall results lack significant difference, but there is a trend towards greater patient satisfaction for BOTOX injection.
It is known that the major complication after hyperhidrosis correction is compensatory sweating. However, there is no mention of evaluation of these parameter by the authors. This way, I think that the results obtained lack strength because this is in fact a significant topic when proposing treatment to patients.
This way, this paper lacks any new scientific findings.
Author Response
"Please see the attachment."

Reviewer 3 Report
This is an interesting prospective interventional comparative study aimed to determine the short term outcome of managing PPH using either intradermal Botox injection (IBI) or endoscopic thoracic sympathectomy (ETS) and evaluate patient satisfaction by the outcome.
It is well designed and described, moreover results are in line with those reported in literature for BTXA vs ETS.
Limits of the study have been honestly reported.
Just a suggestion: authors should report the aim of the study into the introduction section to give the reader the direction of their message from the beginning.
Author Response
"Please see the attachment."

Reviewer 4 Report
Please, allow native English speaker to correct your manuscript to make it better readable - many syntax problems.
You cannot use an acronym like HDSS in the summary without explaining it before.
One issue of interest which should be more clearly worked out: when you acquired your patients and declared these patients the procedure, at this time they did not know into which arm they would fall. I and probably also the readers might be interested to know the fall out of patients after they got informed into which arm they fall. I could imagine that patients allocated to surgery might have decided more frequently to not perform in the study at this moment. Please incorporate this issue in your manuscript.
Author Response
"Please see the attachment."

Round 2
Reviewer 2 Report
Details concerning compensatory hyperhidrosis were added but scarse.
Follow up time (3 months) remains short.
Author Response
"Please see the attachment."
